# OpenReview forum: "Helpful to a Fault: Measuring Illicit Assistance in Multi-Turn, Multilingual LLM Agents"
_ICML.cc/2026/Conference — ICML 2026 regular_

### Official Review · Reviewer_ywD3 · 2026-03-09

**Soundness:** 3
**Presentation:** 3
**Significance:** 3
**Originality:** 3
**Overall Recommendation:** 4
**Confidence:** 4

**Summary:**

This paper studies safety risks in LLM-based agents that interact with users and tools over multiple turns. The authors propose STING, an automated red-teaming framework that simulates adversarial users by decomposing harmful objectives into sequential phases executed through multi-turn interactions. The paper also introduces an evaluation approach that models jailbreak discovery as a time-to-event process and proposes the Restricted Mean Jailbreak Discovery (RMJD) metric. Experiments on the AgentHarm benchmark show that multi-turn attacks significantly increase illicit task completion compared with single-turn prompting and reveal differences in multilingual safety behavior.

**Compliance With Llm Reviewing Policy:**

Affirmed.

**Key Questions For Authors:**

The experiments rely primarily on the AgentHarm benchmark. Have the authors tested STING on other agent evaluation datasets or real-world task environments to verify that the observed vulnerabilities generalize?

STING uses multiple LLM components (strategist, attacker, evaluators). How sensitive are the results to the specific models used for these roles? Would weaker attacker models still produce similar results?

The paper evaluates a simple safety prompt as a mitigation strategy. Have the authors tested other defenses, such as policy filters, tool-access restrictions, or structured reasoning safeguards?

**Limitations:**

Yes. The authors explicitly discuss potential dual-use risks of the framework, acknowledging that techniques for decomposing multi-turn attacks could potentially be misused to improve adversarial strategies. They also note that their work focuses on evaluation rather than providing operational attack instructions and recommend responsible release practices.

**Strengths And Weaknesses:**

This paper addresses an important and timely problem: evaluating safety vulnerabilities in agentic LLM systems that interact with tools and users over multiple turns. As LLM agents become increasingly capable and autonomous, understanding how harmful tasks can emerge through incremental multi-turn interactions is a meaningful research direction. The proposed STING framework provides a structured approach for automated red-teaming by decomposing harmful objectives into phases and coordinating multiple agents that simulate adversarial users. The experimental design is reasonably comprehensive, evaluating several models, multilingual settings, and mitigation strategies. Additionally, the paper introduces an interesting analysis perspective by framing jailbreak discovery as a survival analysis problem, allowing the authors to derive discovery curves and the RMJD metric to quantify how quickly vulnerabilities are exposed. This methodological framing may be useful for future work studying safety evaluation under constrained testing budgets.

However, the paper also has several limitations. First, although the STING framework is clearly described, much of its architecture resembles existing automated red-teaming or jailbreak-generation approaches. The novelty lies mainly in system integration and evaluation methodology rather than fundamentally new algorithms. Second, the experimental evaluation relies heavily on the AgentHarm dataset, which may limit the generality of the conclusions. It remains unclear whether the observed failure modes would persist across other agent benchmarks or real-world deployment environments. Third, the proposed RMJD metric summarizes discovery curves but may not add substantial insight beyond the underlying curves themselves; the paper could further justify why this metric is necessary or how it provides unique interpretability advantages. Finally, while the authors evaluate a simple defense mechanism, the analysis of defenses remains relatively shallow, and stronger mitigation strategies or comparisons with existing safety frameworks would strengthen the practical implications of the work.

---

> ### Author Rebuttal · Authors · 2026-03-31
>
> We thank the reviewer for highlighting the significance of the problem tackled, the comprehensiveness of our experimental design, and our framing of jailbreak discovery as a survival analysis problem.
>
> **Novelty in STING.** While STING builds on existing red-teaming concepts, its design required solving problems absent in chatbot settings: a gated two-stage judge pipeline (Refusal Detector → Phase-Completion Checker) that infers task execution from natural-language responses alone without access to tool-call traces, adaptive attacks over agent-specific failure modes (failed tool calls, missing tool access, hallucinated tool actions; not just refusals), and causally dependent phase sequencing where a single false-positive judge call cascades into full attack failure.
>
> **Why AgentHarm.** AgentHarm is the only available benchmark providing non-binary, graded task-completion scoring for tool-calling agents. In contrast, binary benchmarks (e.g., Agent-SafetyBench) score an agent completing 80% of a harmful workflow identically to one that refuses outright, obscuring the graduated compliance STING is designed to surface. The importance of graded scoring is illustrated by Qwen3-Next: under binary completion (AgentHarm Full Score), STING's improvement appears modest (21.6% → 34.9%), but graded scoring (AgentHarm Partial Score) reveals the agent actually completes 72.6% of harmful workflows (from 32.2%). We agree that testing on real-world environments (e.g., browser-based agents) is a valuable direction, but the barrier is that such environments require fundamentally different grading infrastructure (eg: database state inspection, DOM/screenshot evaluation) rather than the text-based phase-completion checking STING currently uses. We plan to pursue this in future work.
>
> **Effect of the models chosen for each agent in STING.** We replace (i) the judges only with a weaker model (Qwen3-30B-A3B vs. Qwen3-Next-80B-A3B), denoted WE, (ii) both the attacker and judges (WA+WE), and (iii) additionally the strategist (Qwen3.5-27B vs. Gemini3-Pro), denoted WS+WA+WE. Target: Qwen3-Next, Max strategies=5.
>
> | Setting      | English | Ukrainian | Telugu |
> | ------------ | ------: | --------: | -----: |
> | Original     |   67.9% |     70.1% |  71.1% |
> | WE      |  70.7% | 72.4% |      70.1% |
> | WA + WE      |   67.8% |     70.5% |      72.3% |
> | WS + WA + WE |   55.7% |     60.9% |  59.8% |
>
> Weakening judges alone (WE) or both attacker and judges (WA+WE) yields nearly identical AHS, but weakening the strategist has a large impact (~12pp drop), suggesting that phase decomposition quality is the critical bottleneck, not attacker or judge strength.
>
> **RMJD.** RMJD is not intended to replace discovery curves or produce a different scientific conclusion in every setting; it provides a compact, budget-aware summary that enables comparison across methods and especially across different interaction budgets. For instance, in our English results, Qwen3-Next reaches 59.1% ASR in 2 strategies while Gemini reaches 59.4% in 10: nearly identical ASR, yet RMJD (0.99 vs 4.99) reveals Qwen3-Next exposes jailbreaks 5× faster. This distinction is invisible to ASR but critical for resource allocation and fair cross-work comparison as the field matures.
>
> **Defense evaluation.** We additionally evaluate OpenAI's Omni-Moderation API as a filter on both attacker messages and target responses. The following results depict the relative decrease in AgentHarm Scores:
>
> | Setting | English | Chinese | French | Ukrainian | Hindi |  Urdu | Telugu |
> | ------- | ------: | ------: | -----: | --------: | ----: | ----: | -----: |
> | Benign  |    0.0% |    0.0% |   0.0% |     -1.1% |  0.0% | -0.9% |   0.0% |
> | STING   |  -12.5% |  -12.0% | -17.2% |    -10.8% | -7.7% | -7.1% |  -2.4% |
>
> We also evaluated LlamaGuard3, but it rejected 47.2% of benign prompts (Appendix A.7.2), making it impractical. An emerging pattern across all content-level filters (PromptGuard, Moderation API, LlamaGuard) is that they consistently underperform the SafetyPrompt (Table 5): STING's phased decomposition makes individual turns appear benign, which evades per-turn content classifiers. System-level defenses that reason about cumulative intent across turns appear more promising.

---

### Official Review · Reviewer_6RMp · 2026-03-11

**Soundness:** 2
**Presentation:** 3
**Significance:** 2
**Originality:** 2
**Overall Recommendation:** 4
**Confidence:** 4

**Summary:**

This paper studies misuse evaluation for tool-using LLM agents in multi-turn settings.
It introduces STING, where a strategist decomposes a harmful goal into phases, an attacker interacts with the target agent over multiple turns, and two judge modules detect refusals and phase completion.
The paper evaluates STING on AgentHarm across multiple models and languages, compares against single-turn prompting and X-Teaming, and proposes an analysis framework based on time-to-first-jailbreak, including discovery curves, hazard ratios, and RMJD.
The main empirical claim is that multi-turn red-teaming substantially increases harmful-task completion relative to single-turn prompting, while multilingual effects are weaker than prior chatbot-focused work suggests.

**Compliance With Llm Reviewing Policy:**

Affirmed.

**Final Justification:**

My concerns have been adequately addressed.

**Key Questions For Authors:**

1. The multilingual conclusions seem potentially confounded by generating plans in English and reusing them across all attack languages. Can the authors show results when the strategist also plans directly in each target language? A convincing control here would substantially strengthen the multilingual claim.

2. How sensitive are the reported ASR / RMJD conclusions to judge errors, especially given the refusal judge’s reported recall of 0.71 on a relatively small validation set? I would like to see a larger human audit and an analysis of whether model rankings or language conclusions change after adjudication.

3. The comparison to X-Teaming is somewhat hard to interpret because the paper explicitly notes that ASR is not directly comparable, and X-Teaming was designed for chat rather than tool-using agents. Can the authors provide a stronger agent-specific baseline or a more controlled adaptation?

4. The defense study is limited to Qwen3-Next. Do the main conclusions about prompt filtering versus safety prompting generalize to other target models? This would affect the practical significance of the defense findings.

**Limitations:**

yes

**Strengths And Weaknesses:**

1. The paper addresses an important problem and the proposed multi-turn setup is more realistic than standard single-turn agent misuse evaluation. The experimental study is reasonably broad, and the paper includes useful ablations on reasoning effort, tool hints, and lightweight defenses.

2. My main concern is that the central empirical conclusions are not as cleanly supported as the paper suggests. The multilingual claims are confounded because attack plans are generated in English and then reused for all languages, which makes it hard to know whether the results reflect genuine multilingual safety behavior or artifacts of an English-centric attack pipeline. In addition, the evaluation relies heavily on automated judges, yet the reported human validation is limited and the refusal judge has only moderate recall, which could materially affect the reported jailbreak dynamics.

3. The novelty and practical significance feel more incremental than the paper claims. STING combines known ingredients—decomposition, adaptive multi-turn attacks, judge-based evaluation, and AgentHarm-style grading—into an agent setting, but the conceptual advance over recent multi-turn jailbreak and agent-safety work is modest. The new analysis layer is interesting, but it is not yet clear that RMJD or the hazard-based analysis changes conclusions in a meaningful way beyond more standard attack-success summaries. Combined with limited defense evaluation and some fairness issues in the setup, this leaves me unconvinced that the contribution is strong enough for acceptance in its current form.

---

> ### Author Rebuttal · Authors · 2026-03-31
>
> We thank the reviewer for the constructive feedback, recognition of the problem's significance, and our experimental design.
>
> **Native-Language strategies.** We ran STING with strategies generated natively in the target language across Qwen3-Next and Gemini3-Flash on French, Hindi, and Telugu:
>
> |Model|English|French|Hindi|Telugu|
> |----|---:|---:|---:|---:|
> |Qwen3-Next|67.9|68.7 (67.1)|71.2 (73.4)|69.0 (71.1)|
> | Gemini3-Flash |47.0|47.1 (46.3)|47.9 (48.7)|44.5 (46.6)|
>
> Values in brackets are AgentHarm Scores (%) with English strategies (our default setup). Native-language strategies yield AHS within ±2.2pp of English strategies, reinforcing our finding that non-English languages do not amplify jailbreak success in agentic settings, in contrast to chatbot settings.
>
> **Expanded judge validation.** We expanded validation to 570 samples (from 83) for the refusal judge and 260 (from 56) for the intent checker, now covering all 7 languages. Updated precision/recall: refusal judge 0.98/0.93, intent checker 0.99/0.94, substantially improved over the initial validation on two languages.
>
> **Sensitivity of conclusions to judge errors.** We ran experiments replacing the judges with a weaker, more error-prone model (Qwen3-30B-A3B vs. Qwen3-Next-80B-A3B; Max strategies=5, target: Qwen3-Next). Judge-independent AHS values (%) remain stable across judge settings (ASR in brackets):
>
> |Setting|English|Ukrainian|Telugu
> |---|---:|---:|---:|
> |Original|67.9 (75.0)|70.1 (71.9)|71.1(65.3)|
> |Small Judge|70.7 (73.3)| 72.4 (67.6)|70.1(72.2)|
>
> AHS (within 2.8pp) and ASR (with more noise, as it depends directly on judge decisions) both reinforce that non-English languages are not necessarily easier to jailbreak. Furthermore, our core results (Table 1, Figure 3) are based on AHS, which uses rule-based grading functions that verify tool calls, ordering, arguments, and outputs, entirely independent of STING's LLM judges. The model rankings and language conclusions inferred using AHS, therefore, do not depend on judge accuracy.
>
> **Comparison to X-Teaming.** We agree that X-Teaming is not a perfect agent-native baseline, but it is the closest adaptive multi-turn red-teaming framework we are aware of, and we adapt it to the same tool environment as STING for a controlled comparison. Also, the methods are compared using AHS (not ASR) precisely because it is comparable across methods.
>
> **Defense study extended to GPT-5.1 and Gemini 3 Flash.** Results below (relative AHS change (%) vs. no-defense baseline) confirm the same pattern across models (PG=PromptGuard, SP=Safety Prompt):
>
> |Setting|English (PG/SP)|Ukrainian (PG/SP)|Telugu (PG/SP)|
> |---|---:|---:|---:|
> |GPT-Benign|-1.5/-2.7|0.0/-1.9|0.0/-5.4|
> | Gemini-Benign|-1.5/-2.0|0.0/-5.9|0.0/-0.6|
> |GPT-STING| -0.3/-13.8|-1.1/-16.8| -1.5/-25.7 |
> |Gemini-STING|-1.2/-41.1|-0.8/-44.1|0.0/-47.4|
>
> The pattern from Qwen3-Next generalizes: PromptGuard provides minimal defense, while SafetyPrompt yields substantial harmful-task reductions with modest benign impact.
>
> **Novelty.** We respectfully disagree with the characterization as incremental.
>
> (1) *The analysis framework changes conclusions.* RMJD solves a concrete comparability problem: Qwen3-Next reaches 59.1% ASR in 2 strategies, while Gemini reaches 59.4% in 10. These models have a nearly identical ASR, but RMJD (0.99 vs 4.99) reveals Qwen3-Next exposes jailbreaks 5x faster. The Cox model reveals similarly concrete insights: for example, Telugu, the lowest-resource language in our study, hazard ratio confidence intervals are either below 1 or contain 1 across all models (Table 2), formally confirming that it is never easier to jailbreak than English.
>
> (2) *Non-trivial agentic design challenges.* STING is not a straightforward transfer of chatbot attacks. In agentic settings, harmful actions manifest through tool calls that the attacker never observes, so judges must determine phase completion from natural-language responses alone, and distinguish genuine tool execution from parametric confabulation. Furthermore, the adaptive feedback loop must handle agent-specific failure modes absent in chatbot settings (e.g., failed tool calls, tool-access refusals, unexpected tool outputs, hallucinated actions), each requiring different attack adaptation strategies. Finally, phases are causally dependent, so a false positive on any phase cascades into failure on all subsequent ones, making the closed-loop judge design critical. These challenges are unique to the agentic setting and required a novel design.
>
> (3) *First multilingual agentic misuse study.* No prior work has conducted multi-turn adversarial red-teaming of tool-using agents across multiple languages. Our finding that lower-resource languages do not amplify attack success diverges from chatbot patterns, with a direct practical implication: LLM agents may be safer to deploy multilingually than chatbots. We will also release our multilingual tool environment and evaluation suite upon acceptance.

---

> > ### Author Rebuttal · Reviewer_6RMp · 2026-04-03
> >
> > Thanks, I raised my score.

---

> > > ### Author Response · Authors · 2026-04-08
> > >
> > > We thank Reviewer 6RMp for engaging thoroughly with our rebuttal and for raising the score. The points raised on native-language strategies, limited human validation, judge sensitivity, and defense generalization pushed us to produce stronger empirical evidence for our core claims and substantially enhance the rigor of our work. We will incorporate all the additional results into the camera-ready version, if accepted.

---

### Official Review · Reviewer_AWpH · 2026-03-11

**Soundness:** 4
**Presentation:** 3
**Significance:** 3
**Originality:** 2
**Overall Recommendation:** 4
**Confidence:** 3

**Summary:**

LLM agents are equipped with advanced tools and strong memory and can formulate workflows. However, these powerful tools might lead to illicit use and misuse scenarios. Existing benchmarks have addressed this problem by using single-prompt instructions to measure harmful behaviors. While this line of research has greatly advanced our understanding of agents, a gap still exists between single-prompt, single-language settings and multi-turn, multilingual settings, the latter being more practical in real-world interactions. To address this gap, the authors propose a sequential testing framework for illicit N-step goal execution, named STING, focusing on malicious and power-misuse scenarios of agents in multi-turn, multilingual settings. They designed four coordinated agents — a strategist, an attacker, and two judges — to test illicit task-completion probabilities. Results show that STING achieves up to 107.1% higher illicit task-completion rates than predominant single-prompt instructions. They also found that agents in lower-resource language scenarios do not necessarily exhibit increased jailbreak success probability, which differs from previous chatbot findings.

**Compliance With Llm Reviewing Policy:**

Affirmed.

**Final Justification:**

Thanks, the authors fully solved my concern.

**Key Questions For Authors:**

1: What is the basis for problem separation?
2: Do different languages share the same phases?
3: Which setting is more susceptible to jailbreaking and why?

**Limitations:**

No limitations and future works are included in the discussion.

**Strengths And Weaknesses:**

Strengths:
•The multi-turn scenario breaks down the general jailbreak problem into several phases, which can significantly increase the difficulty for the target agent to recognize malicious behavior and thereby improve the attack success rate.

•The idea of comparison across multilingual settings is interesting. Different languages have different information densities, which might also influence the attack rate. For example, languages with lower information density might mislead the target agent, making it harder to understand and recognize malicious behavior.

•Extensive experiments are conducted, and ablation studies are performed on nearly all important variables and parameters, ensuring strict experimental control and thorough analysis.

Weaknesses:
•What is the basis for problem separation? I think the separation of phases is very important, so how to determine these separations and evaluate the quality of the separation should be clearly mentioned and addressed.

•Do different languages share the same phases? The information density varies across languages, and the length of prompts might also be a contributing factor to the results. In Table 2, the authors claim that language has a limited effect on jailbreak success, but I think the length of prompts for each phase may differ across languages. For lower-resource languages, more phases might be needed to achieve a successful jailbreak.

•The authors mention the difference between chatbot settings and tool-use settings. However, in the experiments, the baseline adopts the chatbot setting while the proposed method operates under the tool-use setting. This difference may require further explanation — specifically, which setting is more susceptible to jailbreaking and why?

---

> ### Author Rebuttal · Authors · 2026-03-31
>
> We thank the reviewer for appreciating the increased realism of multi-turn, multilingual attacks and the comprehensiveness of our experimental design.
>
> **Basis for phase separation.** Phase decomposition is handled by the Strategist agent, which is instructed to decompose the harmful intent into "clear, atomic, and executable steps" (Appendix A.2.1, lines 616–618). To quantify separation quality, for each strategy, we consider all pairs of phases and measure the fraction that invoke entirely different sets of tools, where higher values indicate cleaner separation because each phase targets a distinct part of the task. We average this across all behaviors and strategies (176 behaviors, up to 5 strategies each). We compare a strong strategist (Gemini3-Pro) against a weaker one (Qwen3.5-27B), with Qwen3-30B-A3B as attacker/evaluator and Qwen3-Next as target (Max strategies=5).
>
> Phase separation (% non-overlapping tool pairs):
>
> | Strategist    | English | Ukrainian | Telugu |
> |---------------|--------:|----------:|-------:|
> | Gemini3-Pro   |   63.2% |     55.7% |  53.5% |
> | Qwen3.5-27B   |   53.3% |     42.9% |  42.1% |
>
> Despite producing a similar number of phases per strategy (Gemini: 3.36 vs Qwen: 3.19), Gemini3-Pro achieves ~20% more non-overlapping phase executions on average (57.6% vs 46.0%), which directly translates to downstream performance (AgentHarm scores):
>
> | Strategist    | English | Ukrainian | Telugu |
> |---------------|--------:|----------:|-------:|
> | Gemini3-Pro   |   67.8% |     70.5% |  72.3% |
> | Qwen3.5-27B   |   55.7% |     60.9% |  59.8% |
>
> Across languages, Gemini3-Pro yields a higher average AHS (70.8% vs. 58.8%). We view this as evidence consistent with cleaner phase separation improving downstream attack effectiveness, and we will clarify this decomposition criterion more explicitly in the paper.
>
> **Phases across different languages.** In our setup, all languages share the same underlying phase plan in English by design, which keeps the attack structure fixed and lets us vary only the language of interaction. If each language had an independently generated plan, then the phase decomposition itself could differ across languages, making the cross-language comparison less controlled because differences could come from both language and strategy design. To examine the reviewer’s point about information density, we computed token count ratios relative to English and semantic similarity between same-phase attacks across languages (embedder: intfloat/multilingual-e5-small; strategist: Gemini3-Pro; attacker/target/evaluator: Qwen3-Next):
>
> | Language  | Token ratio | Similarity |
> |-----------|------------:|-----------:|
> | Chinese   |        0.98 |      0.727 |
> | French    |        1.32 |      0.751 |
> | Ukrainian |        1.85 |      0.734 |
> | Hindi     |        2.94 |      0.696 |
> | Urdu      |        2.32 |      0.656 |
> | Telugu    |        4.97 |      0.509 |
>
> We observe a strong negative correlation (Pearson's r = −0.94): attack semantics diverge more from English as the language becomes less represented in the tokenizer. This is an interesting finding that complements our main result. Despite this semantic divergence, lower-resource languages do not yield consistently higher jailbreak success (Table 2, Figure 3), suggesting that the divergence does not systematically help bypass safety mechanisms in agentic settings.
>
> **Chatbot vs. tool-use setting.** We clarify that X-Teaming is not run in a chatbot setting in our experiments; the target agent has access to the same tool environment as in STING (Section 5.1, lines 260–263). The comparison is therefore not between “chatbot target” and “tool-use target,” but between a chat-oriented adaptive attack family and our agent-specific phased attack framework, both evaluated against the same tool-using target. The reason X-Teaming underperforms is that it was designed to elicit harmful *text* responses, not to drive multi-step *tool execution*. STING addresses this through its phased sub-task decomposition, operational specificity instructions (e.g., "Shift from Intent to Procedure," Appendix A.2.2), and 2-staged evaluation, which are designed to get each phase’s sub-task done.
>
> **Limitations.** We note that limitations and future directions are mentioned in Section 8 (Conclusion), where we identify extending STING to stronger adaptive adversaries and developing defenses tailored to multi-turn tool use as promising directions.

---

> > ### Author Rebuttal · Reviewer_AWpH · 2026-04-03
> >
> > I thank the authors for their response. I acknowledge the contribution of this work and will maintain my score of weak accept.

---

> > > ### Author Response · Authors · 2026-04-08
> > >
> > > We thank Reviewer AWpH for the constructive feedback and for acknowledging the contribution of our work. The questions raised, particularly on the phase separation quality, were really valuable as these led us to formalize a metric that offers additional insights into the quality of generated strategies. We will incorporate these additional analyses into the camera-ready version, if accepted.

---

### Official Review · Reviewer_qJH1 · 2026-03-15

**Soundness:** 3
**Presentation:** 2
**Significance:** 2
**Originality:** 2
**Overall Recommendation:** 4
**Confidence:** 4

**Summary:**

This paper introduces STING, an automated multi-turn red-teaming framework for tool-using LLM agents. A strategist decomposes harmful intents into phased attack plans, an attacker executes them conversationally against a target agent, and judge agents provide feedback. The paper also proposes a survival-analysis framework (time-to-first-jailbreak, Kaplan-Meier discovery curves, RMJD metric, Cox hazard ratios) and evaluates across 44 AgentHarm scenarios, 5 target models, and 7 languages. Key findings: multi-turn attacks substantially outperform single-turn prompting; non-English languages don't consistently increase jailbreak success (unlike chatbot settings); reasoning effort has non-monotonic effects on safety.

**Compliance With Llm Reviewing Policy:**

Affirmed.

**Final Justification:**

Thanks, the response cleared my concerns.

**Key Questions For Authors:**

See weakness

**Limitations:**

Yes, mostly.

**Strengths And Weaknesses:**

Strengths:

The gap this paper targets is real and timely. Existing agent safety benchmarks like AgentHarm and OS-Harm only test single-turn malicious instructions, so we don't really know what happens when an adversary can adapt over multiple turns. The +107% AHS jump for Qwen3-Next in Table 1 makes the case clear — single-turn evaluations are missing a lot. The survival-analysis framework in Section 4 is probably the most interesting methodological piece. Importing Kaplan-Meier estimators and Cox proportional hazards into red-teaming evaluation is a smart move that gives you cost-aware comparisons rather than just raw success rates. I can see this being adopted by other papers regardless of whether they use STING itself.

Weakness:

The entire pipeline depends on LLM judges for refusal detection and phase completion, but the human validation is quite thin — 83 samples for the refusal judge and 56 for the intent checker. The refusal judge has 0.71 recall, which means it's missing roughly 30% of actual refusals. This could systematically inflate the attack success numbers. More importantly, this validation was only done on Urdu and Telugu conversations, but the judges are used across all seven languages. I'd want to see validation on a broader sample before fully trusting the multilingual results.

There's a confound in the multilingual setup that deserves more attention. Attack plans are generated in English by Gemini 3 Pro and then translated into the target language (Appendix A.4). So non-English attacks are really translated English strategies, not natively-conceived multilingual attacks. This could partly explain why non-English doesn't help — the attack quality may simply degrade through translation rather than this being a fundamental property of agentic settings. The paper should acknowledge this more directly.

The related work section overlooks relevant prior work on multi-turn agent safety. In particular, HaicoSystem (Zhou et al., 2024) already evaluates agent safety in multi-turn, tool-augmented settings with simulated social interactions. The claim in Section 2.2 that agentic safety benchmarks "typically issue a single-turn malicious instruction" should be qualified in light of this. The paper should discuss how STING's approach (adversarial phased decomposition with adaptive feedback) differs from HaicoSystem's simulation-based evaluation.

---

> ### Author Rebuttal · Authors · 2026-03-31
>
> We thank the reviewer for the constructive feedback and for recognizing the timeliness of the problem and the value of our survival-analysis framework.
>
> **Expanded human validation of judges.** During the rebuttal period, we expanded validation to 570 samples (from 83) for the refusal judge and 260 (from 56) for the intent checker, now covering all 7 languages. Updated precision/recall: refusal judge 0.98/0.93, intent checker 0.99/0.94, substantially improved over the initial validation on two languages. Furthermore, the two-stage pipeline design (Figure 1) mitigates failures in refusal detection. As a result, a missed refusal can be independently caught by the intent checker before any phase is marked complete.
>
> Finally, our core results comparing methods (Table 1) and languages (Figure 3) use AgentHarm Scores (AHS), computed via AgentHarm's rule-based grading functions that are entirely independent of STING's LLM judges. The key insight that non-English languages are not more susceptible to jailbreaks in agentic settings, therefore, rests on judge-independent evaluation. The high correlation between ASR and AHS (Pearson's r = 0.96, Line 349) further validates the robustness of our two-stage evaluation.
>
> **Clarification on the multilingual setup.** In STING, plans are generated in English to control for strategy-generation quality across languages. Then, the attacker agent *natively generates* its attacks in the target language (Section A.2.2, lines 694–695, 741). The non-English attacks are therefore not "translated English strategies". The attacker produces its utterances directly in the attack language, guided by but not translating the English plan.
>
> To verify whether native-language strategies would change our conclusions, we ran STING with strategies generated natively in the target language across Qwen3-Next and Gemini3-Flash on French, Hindi, and Telugu:
>
> | Model         | English |        French |         Hindi |        Telugu |
> | ------------- | ------: | ------------: | ------------: | ------------: |
> | Qwen3-Next    |   67.9% | 68.7% (67.1%) | 71.2% (73.4%) | 69.0% (71.1%) |
> | Gemini3-Flash |   47.0% | 47.1% (46.3%) | 47.9% (48.7%) | 44.5% (46.6%) |
>
> Values in brackets are AHS with English strategies (our default setup). Native-language strategies yield AHS within ±2.2pp of English-plan strategies across all conditions, reinforcing our finding that non-English languages do not amplify jailbreak success in agentic settings, in contrast to chatbot settings (Deng et al., 2023).
>
> **HaicoSystem in related work.** We thank the reviewer for raising this point and agree that HaicoSystem (Zhou et al., 2024) is a relevant prior work that should be discussed. We will revise Section 2.2 to qualify the single-turn claim accordingly and identify the distinctions between the two works: HaicoSystem is a simulation sandbox where an LLM-simulated user role-plays a scenario with an AI agent, and safety evaluators assess the resulting trajectory. STING, by contrast, is an adversarial red-teaming method that actively maximizes illicit-task completion through phased goal decomposition and closed-loop adaptive feedback, making STING explicitly optimized for adversarial success, whereas HaicoSystem is a simulation framework for evaluating emergent safety risks in multi-turn interactions.

---

> > ### Author Rebuttal · Reviewer_qJH1 · 2026-04-02
> >
> > Thanks, I raised my score.

---

> > > ### Author Response · Authors · 2026-04-08
> > >
> > > We thank Reviewer qJH1 for engaging thoughtfully with our rebuttal and for raising the score. The additions directly strengthened the empirical rigor and overall quality of our work. We will incorporate all discussed revisions: expanded judge validation, native-language strategy experiments, and the HaicoSystem discussion, into the camera-ready version, if accepted.

---

### Decision · Program_Chairs · 2026-04-30

**Decision:**

Accept (regular)

**Comment:**

This paper introduces STING, a multi-turn red-teaming framework for evaluating tool-using LLM agents, along with a survival-analysis-based evaluation methodology for measuring jailbreak dynamics.

The reviewers provided consistently positive scores (all Weak Accepts). They agreed that agent safety in multi-turn, multilingual settings is an important and timely problem, and appreciated both the framework design and the proposed analysis tools. The main concerns focused on judge reliability, potential confounds in the multilingual setup, and questions about novelty and evaluation breadth.

During rebuttal, the authors provided strong and targeted clarifications. They substantially expanded human validation of the judges, added native-language strategy experiments to address multilingual concerns, and included additional analyses on judge sensitivity and defense generalization. Reviewers acknowledged that these additions resolved their concerns and maintained or raised their scores.

While some limitations remain—such as reliance on a single benchmark and moderate novelty as a system integration—the paper offers a solid and practical contribution to agent safety evaluation. Given the strengthened empirical support and clear positive reviewer consensus, I recommend Accept.